# Pediatric Low-Grade Glioma Surgery with Sodium Fluorescein: Efficient Localization for Removal and Association with Intraoperative Pathological Sampling

**DOI:** 10.3390/diagnostics12122927

**Published:** 2022-11-23

**Authors:** Camilla de Laurentis, Pierre Aurélien Beuriat, Fred Bteich, Carmine Mottolese, Alexandru Szathmari, Matthieu Vinchon, Federico Di Rocco

**Affiliations:** 1Pediatric Neurosurgery Unit, Hôpital Femme Mère Enfant, 69003 Lyon, France; 2Department of Medicine and Surgery, Università degli Studi di Milano Bicocca, 20126 Milano, Italy; 3Medicine and Surgery Department, Université Claude Bernard Lyon I, 69008 Lyon, France

**Keywords:** sodium fluorescein, yellow 560 nm, pediatric neurosurgery, neuro-oncology, low-grade gliomas, astrocytoma, ganglioglioma

## Abstract

Low-grade gliomas are among the most common CNS lesions in pediatrics and surgery is often the first-line treatment. Intraoperative tools have been developed to maximize the results of surgery, and in particular dyes such as sodium fluorescein (SF) have been investigated in high-grade adult lesions. The use of SF in pediatric low-grade gliomas is still unclear. We retrospectively reviewed 22 pediatric CNS low-grade gliomas operated on with SF from September 2021 to October 2022. A total of 86% of lesions showed SF uptake, which was helpful intraoperatively (confirmation of initial localization of the tumor, or identification of tumor remnants) in 74% of them. The intraoperative fluorescence seems associated with gadolinium enhancement at the preoperative MRI. Interestingly, the extemporaneous pathological sampling (EPS) was informative in every case showing SF uptake, whereas in cases without SF uptake, the EPS was non-informative, although the tissue was later confirmed as pathological. These findings highlight the interest of SF for perioperative diagnosis of tumor tissue and may suggest in which cases the differentiation of tumor–healthy tissue could be especially blurred, posing difficulties for the pathologist.

## 1. Introduction

Low-grade gliomas (LGGs) are among the most common solid neoplasms in children, accounting for about 40% of all tumors of the central nervous system (CNS) in the pediatric age [1,2].

A preliminary diagnosis can be suspected from clinical presentation and then neuroimaging, in particular magnetic resonance imaging (MRI). WHO grade I lesions exclusively include pilocytic astrocytomas, that often present with a solid, gadolinium enhancing nodule next to a cystic cavity [3]. A certain degree of gadolinium enhancement can be also seen in some WHO grade II gliomas, although it is less constant. Their ability to uptake gadolinium, which passes through the blood–brain barrier (BBB) by diffusion, could be explained by an alteration in the barrier itself, more permeable than the one surrounding normal parenchyma. This is normally a characteristic of high grade lesions, but it has been shown that it can be found even if the tumor does not show any sign of malignancy [4].

In recent years, an always-increasing understanding of the molecular biology of LGGs has opened more opportunities for applying targeted therapeutic approaches [5]. Nevertheless, surgery still remains the first-line treatment, as it both provides pathological samples and allows a rapid reduction in tumor volume, which should always be a maximal safe extent of resection (EOR) whenever possible [6]. Indeed, in some cases of LGGs in a favorable location, complete resection is considered curative, and EOR may still be a key predictor of clinical outcome [7].

However, the surgical procedure may be very challenging in infiltrating LGG, in particular in localizing the tumor–healthy interface at the beginning of surgery and remnants at the end. In recent decades, a wide range of different tools have been proposed to maximize the results of surgery, both in adult and pediatric oncological neurosurgery. We have seen the development of awake surgery [8,9], accurate preoperative and intraoperative imaging (from MRI to ultrasound), neuronavigation, intraoperative neuromonitoring, and fluorophores.

Among the latter, the most used ones are 5-aminolevulenic acid (5-ALA) and sodium fluorescein (SF). 5-ALA is a precursor of protoporphyrin IX (PpIX) with an enzymatic mechanism of action: its metabolism leads to the accumulation of porphyrins in cells with high turnover rate, and they appear as a pinkish fluorescence that can be enhanced when seen under a fluorescent 400-nm UV light [10]. Because of its nature, 5-ALA is often considered a tumor-specific fluorescent dye that works better in high-grade gliomas [11]. A certain number of reports on the LGGs [12,13], and more specifically on LGGs in pediatrics [14,15,16], can be found in the literature, generally showing absent or inconsistent fluorescence.

On the other hand, sodium fluorescein, the sodium salt of fluorescein, accumulates in areas with an altered BBB through simple diffusion from the intravascular space to the extracellular one. This mechanism of action is reminiscent of that of gadolinium, therefore seemingly promising not only in high-grade gliomas [17], but also in low-grade enhancing lesions like some LGGs [18,19,20], and especially in pediatric LGGs [21,22].

For this reason, we report the experience of our institution with SF-assisted surgery in children operated on for a LGGs, evaluating its utility, relationship with gadolinium enhancement, and possible association with anatomopathological findings, to evaluate the possible role of SF in the landscape of diagnosis and maximal treatment of pediatric LLGs.

## 2. Materials and Methods

### 2.1. Patient Population

We retrospectively reviewed the charts, the radiological findings, and intraoperative images of children operated for low grade gliomas (WHO grade I and II) at our Institution from September 2021 to October 2022 (IRB 00011687, 2022/46). We extrapolated demographic and anatomopathological data, radiological characteristics, intraoperative data (SF administered dose, SF injection time, incision time, time on tumor, registered adverse reactions), and surgeon’s experience (intraoperative intensity of fluorescence subjectively graded by the first operator as +++ or bright, ++ or moderate, + or modest, - or absent, at the end of surgery; utility of SF in differentiating between lesion and healthy tissue; and utility for initial localization of the tumor or the identification of tumor remnants).

### 2.2. Surgical Procedure

At the end of anesthesia induction, the patient was injected intravenously with a dose of 5 mg/kg of SF (Fluorescéine sodique faure 10%, SERB SA, Bruxelles, Belgium). Parental informed consent was obtained.

We operated with the aid of a microscope (KINEVO 900, Zeiss Meditec, Oberkochen, Germany) equipped with a filter specific to SF visualization (YELLOW 560 nm). At tumor exposition, the first filter switch was made, then the surgeon was given the freedom to switch from white light to the YELLOW 560 nm filter at any time during the whole procedure. We took a photo respectively under YELLOW 560 nm and under white light at tumor exposition, then photos and videos were freely taken throughout surgery.

When it seemed useful, the surgeon asked for an extemporaneous pathological sampling (EPS) of the tumor. At the end of the intervention, the surgeon rated the intraoperative fluorescence as a semi-quantitative variable (+++, ++, +, or -) and evaluated the utility of SF in localizing the tumor or its remnants.

### 2.3. Image Analysis

For each patient, a representative T1-pre-contrast and a T1-post-contrast image were selected for subjective image analysis by two of the authors. Contrast enhancement (CE) patterns were subdivided as (A) mostly diffuse, with possible (a) small cyst(s); (B) inhomogeneous, with areas of CE at the periphery (ring enhancement) or within the tumor, and large non-enhancing areas; (C) enhancing nodule with a single non-enhancing large cyst. The degree of the area showing CE was rated as +++, ++, +, or -.

Photos taken at tumor exposition (under white light and under a YELLOW 560 nm filter) and stored in the surgical microscope were retrieved and analyzed through an open-source Java image processing platform, ImageJ (Version 1.53, National Institute of Health, Bethesda, MD, USA) as already suggested elsewhere [23]. The platform returned for each ROI a color analysis quantifying the content of each color, and we registered the green quantities for the tumor region both under white light and under the filter. We calculated the increase in green quantity from white light to filter vision both in tumors and in healthy tissue (Figure 1).

### 2.4. Pathological Analysis

In the cases in which the surgeon judged it useful, an EPS was performed, analyzed as a frozen section. A definitive pathological analysis was performed in every case, classifying the results according to the 2021 WHO Classification of the Tumors of the Central Nervous System [24].

### 2.5. Statistical Analysis

We described our series through standard descriptive statistics. We applied the Fisher exact test for binary variables. A threshold of *p* < 0.05 was set for statistical significance.

## 3. Results

### 3.1. Patient Population and Surgical Procedure

Our study included 22 patients presenting a CNS low-grade glioma operated on from September 2021 to October 2022 with the aid of SF. Their ages ranged from 19 months to 17 years and 7 months, and their weight from 14 to 67 kg. The lesion was supratentorial in 12 cases, infratentorial in 7, and intramedullary in 3.

All patients received a dose of 5 mg/kg of SF; in the majority of cases, the injection was given at the end of induction (15/22), in 4/22 at incision or after that, and in 3/22 cases we had no data about injection time.

Neither immediate nor late adverse reactions were registered. In every case, from about the end of surgery and in the following hours, a yellowish discoloration of urine was observed, which was anyway self-limiting and without any sequelae. No alterations at intraoperative and postoperative exams were noted.

From the pathological point of view, a total of 15 pilocytic astrocytomas (PA, of which 11 were with the KIAA1549-BRAF transcript and 4 without), 4 gangliogliomas (GG, 2 with BRAF v600e and 2 without), and 1 each of glial tumor of the optic pathways KIAA1549-BRAF, epileptogenic oligoid tumor CD34+, and angiocentric glioma BRAF+ were included. All the included lesions were grade I according to the 2021 WHO Classification of the Tumors of Central Nervous System [24].

The intensity of intraoperative fluorescence declined on a histological basis, completed with genetic analyses, is shown in Figure 2.

In total, 19/22 (86%) of the lesions showed some degree of fluorescence, and it turned out as moderate or bright in 16/19 (84%).

Because of the limited number of cases for each histology, no association can be hypothesized between a certain pathological result and a certain reproducible intensity of fluorescence. Exclusively for pilocytic astrocytomas, independently from their molecular characteristics (*n* = 14), we can observe in our series a quite constant moderate–intense SF uptake (12/14, 86%).

### 3.2. Image Analysis

An image analysis was performed for each case, both under white light and under the YELLOW 560 nm filter. Green difference under the YELLOW 560 nm filter seems to be a good objective counterpart of the subjective precepted discrimination between healthy and tumor tissue, as published by another group [23] and as previously reported at our Institution (article in the publishing process).

We then hypothesized that the objective intensity of intraoperative fluorescence could be described by:∆ tumor(1)
i.e., the difference (or delta, ∆) between the green value of the tumor under the filter and the green value under white light, or by:∆ tumor − ∆ healthy(2)
i.e., the difference (or delta, ∆) between the green intensity increase (from white light to filter) in tumor and the green intensity in healthy tissue, or by:∆ tumor − ∆ healthy/Δ healthy(3)
i.e., Equation (2) corrected for Δ healthy tissue.

The results of these analyses can be found in Table 1. A total of 19 images were available for the study. In 11/19 cases the subjective perception was intense, the values from (1) ranged from 1569 to 210,597, the ones from (2) were included between −32,517 and 184,872, and the ones from (3) from −132,771 to 3693. The values obtained from the cases whose fluorescence was classified as moderate, modest, or absent were all included in the range already covered by cases with intense fluorescence. No clear progression can be seen numerically, through these equations, from intense to low/absent fluorescence.

### 3.3. Intraoperative Fluorescence Relationship with Preoperative MRI Gadolinium Enhancement Pattern and Intensity

As far as it concerns patterns of preoperative MRI gadolinium enhancement, in the overall group of 22 lesions it resulted as follows: six type A (27%), six type B (27%), five type C (23%), and four not enhancing (19%).

The degrees of fluorescence for each pattern are shown in Table 2.

Although the number of cases is very limited, we can observe that in pattern A (diffuse CE) and C (enhancing nodule) intraoperative fluorescence was always present, and in 8 out of 11 cases it was precepted as bright. On the other hand, in pattern B (inhomogeneous contrast enhancement) fluorescence was bright in only one case and absent in another case, while in the others it was rated as moderate or modest. In pattern D (no contrast enhancement), in four out of five cases fluorescence was absent or very limited.

When analyzing the intensity (regardless of pattern) of preoperative MRI gadolinium uptake of the enhancing part of each lesion, we compared intensity at MRI rated as +++, ++, +, - and intensity of intraoperative fluorescence equally rated as +++, ++, +, -. The results are shown in Table 3.

Out of the 18 lesions with CE at preoperative MRI, 17 showed intraoperative fluorescein (94%). Fluorescein seems to have a similar or higher intensity when compared to gadolinium intensity: perfect intensity correspondence was seen in 13/22 cases (59%), but in another 6 cases it resulted brighter than seen at MRI CE (6/22, 27%). In particular, SF uptake was seen in 2/4 lesions not showing preoperative gadolinium enhancement.

The only lesion showing preoperative CE but no fluorescein uptake was a very small ganglioglioma remnant, with small spots of CE (pattern B), so this heterogeneity is probably the reason why fluorescence was not precepted.

### 3.4. Utility of SF in Localizing Pediatric LGGs

The surgeon was asked about the utility of SF in localization, i.e., initial localization and confirmation of lesion tissue (Figure 3), or identification of tumor remnants towards the end of surgery. For this aim, SF turned out to be helpful in 14/19 (74%), and indispensable in 2 of them. In the remaining 5/19 tumors, the lesion was already extremely well differentiated, so confirmation of localization was judged unnecessary.

We compared the utility of SF in localizing tumors in lesions which showed fluorescence to its non-utility in non-fluorescent lesions. The Fisher exact test resulted as *p* = 0.0364 (*p* < 0.05).

### 3.5. Extemporaneous Pathological Sampling (EPS) and SF

An extemporaneous pathological sampling (EPS) was performed in 17 cases. We considered it “informative” when the pathologist could give an immediate, generic pathological diagnosis with reasonable certainty (tumor or not, and glial line or not), while we considered it “non-informative” when they could not determine if the tissue was surely pathological, or when a certain pathological diagnosis over another could not be hypothesized. In every case, a definitive pathological diagnosis could be obtained in the days following surgery, confirming that every sample was effectively tumor tissue.

Interestingly, the extemporaneous pathological sampling (EPS) was informative in every case showing SF uptake (15/17, 88%), whereas in the 2/17 cases (12%) without SF uptake, the EPS was non-informative, although in every case a definitive pathological diagnosis could be obtained in the days following surgery, confirming that every sample was effectively tumor tissue.

## 4. Discussion

### 4.1. Sodium Fluorescein in Pediatric LGG: Considerations, Feasibility, and Safety

Our case series shows quite constant intraoperative fluorescence from SF in pediatric LGGs and enhances its feasibility and capability in assisting intraoperative diagnosis.

Globally, 86% of the lesions included showed some degree of fluorescence (from bright to modest), which can be useful in confirming the localization of the tumor at the beginning of surgery or its remnants at the end.

In the literature, different cases of adverse reactions have been reported, but interestingly they either involve high dosages of SF [25] and/or intrathecal injection [26,27,28]. No serious adverse events have been reported so far in a pediatric population [21,29,30], as shown in our LGGs series. Only a self-limited yellowish discoloration of urine was observed, which has been constantly reported [20,31] and is expected due to the known mechanism of elimination of SF through a renal washout [32].

### 4.2. Association between Fluorescence and Gadolinium Enhancement and Image Analysis

The present case series includes a limited number of patients, so no definitive association can be found between a certain LGG histology and a reproducible intensity of fluorescence. Nevertheless, we can note that for pilocytic astrocytomas, independently from their molecular characteristics (*n* = 14), a quite constant moderate–intense SF uptake has been registered (86%), probably due to the constant presence of a well-defined alteration of the BBB in this type of tumor [4]. This alteration, also found in other pediatric LGGs, can explain the relationship between SF uptake and gadolinium enhancement at the preoperative MRI. This association has repeatedly been reported in the literature [22], in some cases even more precisely in some pathologies than in others, for example in astrocytic tumors [31], and is generally explained by a similar mechanism of action shared by both SF and gadolinium. In fact, they both spread in the extracellular tissue through diffusion, and they accumulate in areas where the BBB is altered. The unperfect matching of the two, with SF being able to spread in areas of absent contrast enhancement [33,34], is usually explained by the different molecular sizes, i.e., SF being slightly smaller, it can also pass into areas where gadolinium cannot.

In our series, 94% of the LGGs with CE at preoperative MRI showed intraoperative SF uptake. A perfect intensity correspondence was seen in 59% cases, but in another 27% it even resulted brighter than seen at MRI CE, and SF uptake was seen in 2/4 lesions not showing preoperative gadolinium enhancement (Table 3). The only case showing preoperative CE but no intraoperative SF uptake was a limited ganglioglioma remnant, with small spots of CE, so this heterogeneity may be the reason why fluorescence was not precepted.

Interestingly, although statistical analyses are limited by the extremely small sample size, when analyzing SF with respect to a gadolinium enhancement pattern, we observed a generally constant fluorescence in cases where gadolinium enhancement was diffuse or focused on a single, enhancing nodule (Table 2). On the one hand, a diffuse CE may suggest an equally diffuse SF uptake, which is precepted as a globally bright color. On the other hand, this tendency could be also explained through a link to a pathological result. In other words, a certain gadolinium pattern is generally seen in a specific pathology (for example a single, enhancing nodule with a large cyst is quite typical of a pilocytic astrocytoma), which can be the actual reason behind a certain degree of fluorescence.

Further studies are required to evaluate all the intervening elements which contribute to a certain intensity of fluorescence and its link to gadolinium enhancement, among which are the time from injection of SF to the evaluation of fluorescence, the pathology itself, and drugs that enhance or blur the alteration in the BBB.

With the aim of objectivating this evaluation of fluorescence, until now always described through subjective evaluations [20,22,23,31], and facilitating comparison to gadolinium enhancement intensity, we tried to describe the objective intensity of intraoperative fluorescence through differences in green values, both under white light and a filter, both in healthy and tumor tissue. A difference in green intensity under the YELLOW 560 nm filter seems a good objective counterpart to the subjective precepted discrimination between healthy and tumor tissue (as published by another group [23] and as previously observed at our Institution, to be reported in a forthcoming article), so we tried a similar evaluation of the difference of green under white light vs. under the filter. However, no trend could be seen, suggesting that an extremely simplified and linear color analysis cannot well describe SF uptake. New insights are welcomed to evaluate other possible image analysis tools, and if possible, to integrate them directly in the surgical microscope as already implemented for vascular neurosurgery [35].

### 4.3. Utility of SF in Localizing Pediatric LGGs and Association with EPS

SF was revealed helpful in 74% of LGGs showing uptake, either in confirming the localization of the altered tissue at tumor exposition or in individuating remnants at the end of surgery. What is extremely interesting is that in two cases SF was judged “indispensable” by the surgeon, i.e., surgery could have been developed or ended in another way without SF assistance. A possible future field of investigation could be the evaluation of SF in combination with other intraoperative tools to further reciprocally augment their efficiency in tumor resection, for example as already reported for 5-ALA associated with intraoperative ultrasound [36].

When analyzing the association with EPS efficacy, we found that EPS was informative in every case showing SF uptake (88%), whereas in the cases (12%) without SF uptake, the EPS was non-informative, although in every case a definitive pathological diagnosis could be obtained in the days following surgery, confirming that every sample was effectively tumor tissue.

These findings globally suggest, on the one hand, that SF can be an extremely useful part not only in tumor removal but also in the macroscopic diagnostic of LGGs. In other words, it can be integrated into the intraoperative immediate pathological evaluation, shared by both the surgeon and the pathologist. On the other hand, the cases without SF uptake which also returned a non-informative EPS may suggest the presence of especially complex situations, in which the limits between healthy and pathological tissue are blurred, and they could be very difficult to define not only for the surgeon but also the pathologist when working without dyes, fixatives, and genetic analysis.

## 5. Conclusions

SF in pediatric LGGs seems feasible, safe, and useful in assisting intraoperative diagnosis. It can be very useful in confirming the localization of the altered tissue at the beginning, or in individuating remnants at the end of surgery, and it can also acquire a role as an essential part of the intraoperative immediate pathological evaluation, shared by both the surgeon and the pathologist.

## Figures and Tables

**Figure 1 diagnostics-12-02927-f001:**
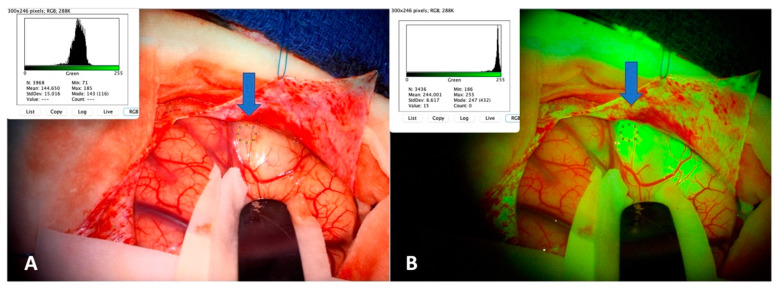
Example of image analysis for tumor tissue. For the selected ROI (blue arrows) the program gives a value of green (**A**) under white light, (**B**) under the YELLOW 560 nm filter. We performed the same analysis for the adjacent healthy tissue, both under white light and YELLOW 560 nm filter.

**Figure 2 diagnostics-12-02927-f002:**
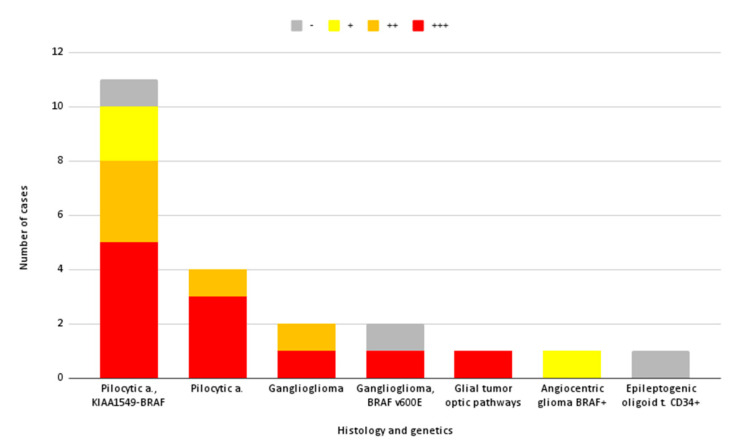
Intensity of intraoperative fluorescence for each histology and molecular characteristics. +++: intense; ++: moderate; +: modest; -: absent fluorescence.

**Figure 3 diagnostics-12-02927-f003:**
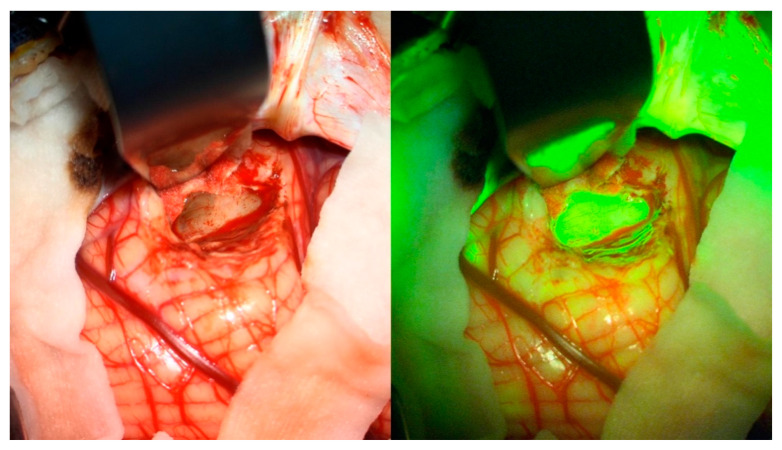
Example of utility in confirming localization. The neuronavigation and the observation using a microscope suggest the surgeon may have reached the tumor (**left**); intraoperative SF fluorescence confirms the localization of altered tissue (**right**).

**Table 1 diagnostics-12-02927-t001:** Comparison of subjective fluorescence perception to numerical differences in green intensity.

Subjective	∆ Tumor	∆ Tumor − ∆ Healthy	∆ Tumor − ∆ Healthy/∆ Healthy
+++, homog	1.569	−32.517	−0.954
+++, homog	5.889	−12.949	−0.687
+++, homog	10.354	14.377	−3.574
+++, homog	68.188	34.958	1.052
+++, homog	69.271	95.757	−3.615
+++, homog	92.411	93.253	−110.752
+++, homog	124.392	125.336	−132.771
+++, homog	170.356	181.507	−16.277
+++, homog	181.021	184.872	−48.006
+++, homog	202.001	158.957	3.693
+++, homog	210.597	−7.739	−0.035
++, homog	−1.514	−11.415	−1.153
++, homog	98.78	108.856	−10.803
++, heterog	129.525	98.389	3.160
++, homog	164.757	142.636	6.448
++, homog	183.429	89.185	0.946
+, homog	46.275	7.413	0.191
-	22.654	−37.171	−0.621
-	48.519	5.938	0.139

+++: bright fluorescence; ++: moderate fluorescence; +: modest fluorescence; -: absent fluorescence.

**Table 2 diagnostics-12-02927-t002:** Pattern of gadolinium enhancement and degree of intraoperative fluorescence.

SF Uptake Pattern MRI	+++	++	+	-
A	4	2		
B	1	3	1	1
C	4	1		
D	1		2	2

+++: bright fluorescence; ++: moderate fluorescence; +: modest fluorescence; -: absent fluorescence.

**Table 3 diagnostics-12-02927-t003:** Preoperative MRI gadolinium enhancement intensity and degree of intraoperative fluorescence.

MRI\SF	+++	++	+	-
+++	7	2		
++	3	3		
+		1	1	1
-	1		1	2

+++: bright fluorescence; ++: moderate fluorescence; +: modest fluorescence; -: absent fluorescence.

## Data Availability

Not applicable.

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
