# Peer review of "Pediatric Low-Grade Glioma Surgery with Sodium Fluorescein: Efficient Localization for Removal and Association with Intraoperative Pathological Sampling"

_diagnostics, 2022, doi:10.3390/diagnostics12122927_

Round 1
Reviewer 1 Report
These authors report on their experience with sodium fluorescein in the resection of pediatric low grade gliomas.
1) The introduction can be shortened.
2) The authors describe a qualitative grading scale. Can they provide more detail on how this was determined? Are there representative images they can show? What was the inter-rater reliability?
3) The authors also measured fluorescence quantitatively. If fluorescence was measured quantitatively, why was a qualitative assessment also included, especially as there seems to be little correlation between the surgeons’ assessments and the measurement?
4) What was the criteria for grading degree of enhancement as +++, ++, or +? Are there representative images? Was quantitative data collected? Who reviewed the images, what was their training? If more than one reviewer was used, what was the inter-rater reliability?
5) I am unsure what the utility is of defining “informative” EPS as providing a definitive diagnosis. It seems that the most useful answer a pathologist could provide is simply “tumor” or “not tumor.”
Reviewer 2 Report
Interesting and well written study regarding Pediatric low grade gliomas surgery with sodium fluorescein, with a focus on efficient localization for removal and association with intraoperative pathological sampling.
Introduction: nothing to concern.
M&M: well explained.
Results: well presented and complete.
Discussion: Just a minor comment - could you please comment about the possibility of combining fluorescence with ultrasonography to increase tumor resection (eg: Della Pepa GM et al. "Dark corridors" in 5-ALA resection of high-grade gliomas: combining fluorescence-guided surgery and contrast-enhanced ultrasonography to better explore the surgical field. J Neurosurg Sci. 2019 Dec;63(6):688-696. doi: 10.23736/S0390-5616.19.04862-8. PMID: 31961118.) ?
Conclusion: effective and concise
Reviewer 3 Report
The authors well explored the efficient localization for removal and association with intraoperative pathological sampling
